

# Crumbling cliffs and intergenerational cohesivity: A new climate praxis model for engaged community action on accelerated coastal change

Katie J. Parsons[1], Florence Halstead[2], Lisa M. Jones[3], Sarah Harris-Smith[4]

[1]Department of Geography and Environment, Loughborough University, Loughborough, UK
[2]School of Education, University of Glasgow, Glasgow, UK
[3]School of Education, University of Hull, Hull, UK
[4]Withernsea High School, Withernsea, UK

*Correspondence to*: Katie J. Parsons (k.j.parsons@lboro.ac.uk)

**Abstract.** Climate change is widely accepted as an existential threat that requires urgent action globally, regionally and locally. Despite the challenge there remains a lack of awareness among many in society regarding the scale of the environmental changes and projected impact(s) on lives and livelihoods. Despite climate change being a prominent topic in politics and activism, broader engagement with the climate crisis in sections of society, particularly in disadvantaged communities remains lower than across society as whole. Part of these issues relate to unequal access to information and limited resources in some communities, which together contributes to a knowledge gap. Moreover, disinformation campaigns, fake news, and biases in media further complicate understanding of the climate crisis across sections of society. Here we report on the INSECURE project, which had the aim to engage a disadvantaged coastal community that is very much on the front line of climate change. The engagement was advanced through creative methodologies and intergenerational dialogues to bridge the gap between climate science, knowledge and public understanding through innovative ways to educate and communicate the issues of climate change. By considering individuals' attitudes, beliefs, cultural backgrounds, and lived experiences, the project seeks to overcome misconceptions and confusion. The results show the importance of knowledge and how knowledge gaps can act as a barrier for individuals in engaging with the climate crisis. The results additionally highlight how employing new and creative communication approaches can empower a disadvantaged coastal community with the understanding necessary to address climate change within their local context(s) and thus ensure that communities can be better prepared and equipped to face the future impacts of climate change.

## 1 Introduction

*'Experience is the only thing that brings knowledge, and the longer you are on earth the more experience you are sure to get'.*

L. Frank Baum (1900), The Wonderful Wizard of Oz





Climate change is an existential threat to society and is a challenge we are failing to address at the scale and speed required in order to meet the recommendations and targets identified by the Intergovernmental Panel on Climate Change (IPCC, 2021). Addressing climate change requires a suite of measures combining mitigation and adaptation, from rapid expansion of renewable energy and electrification through to adoption of more sustainable forms of transport, alongside measures to adapt communities to changes in, for example, sea-level and flood risk (Kulp and Strauss, 2019). However, each of these requires

societal engagement and the systemic adoption of climate mitigation and adaptation policies by governments. In the context of engaging societies in addressing climate change, Jones et al. (2021) describe the journey through an emotional framework, highlighting how those confronting and understanding the scale of the climate change challenge and the scale of the issues at play face feelings of loss and grief, before engaging with taking action. Jones et al. (2021) describe climate change as representing:

*"...the biggest form of (un)imaginable loss to humans; at worst, the threat to life as we know it and our existence. It represents loss of certainty about what the future holds, a loss of confidence that we can simply carry on as we were and the potential for a loss of, most worryingly, of hope for the future."*

However, despite climate change being high on political agendas and with a suite of high-profile activism gaining momentum around the world (e.g. Fisher and Nasrin, 2021), there remains many across society who are still largely unaware of the scale

of the environmental change that is now upon us (Lee et al., 2015; Morote and Hernández, 2022), how human activity has been the direct cause of these changes and, perhaps most importantly, how this is going to impact their, and future generations', lives and livelihoods.

Finding new and creative ways to engage a broad range of diverse, particularly disadvantaged, communities with the climate crisis has never been more crucial, preparing and equipping them for what the future holds. Science and research play a key

role in helping shape the debate on addressing climate change, but there is well known unequal access to knowledge through inequitable educational opportunities, paywalls in scientific journals and limited time and funding for public engagement and science communicators (Dorkenoo et al. 2022; Allam et al., 2002). This results in many across society relying on other forms of information that are often not up to date, unreliable, fake or communicated from bias viewpoints (Hassan et al., 2023; Piatek et al., 2024). Moreover, long-running disinformation campaigns still have resonance and impacts on understanding (Lutzke et

al., 2019) and the rise of "Fake News" and social media inaccuracies have continued to propagate confusion and disengagement from addressing the climate crisis (c.f. Scheufele and Krause, 2019; Boykoff and Boykoff, 2007). As a result, understanding the overall meaning of climate change can vary widely in individuals and across community groups due to these propagating misconceptions, attitudes, culturally-driven beliefs, lived experiences and general confusion of the topic (Otto, 2017). This lack of accurate knowledge is often a significant barrier to a range of communities and sub-sections of society engaging with

the climate crisis (Sarabi et al., 2020; Lorenzoni et al., 2007).



Access to information and opportunity varies depending on both age and socio-economic drivers. Many young people, particularly from disadvantaged backgrounds, are left out of conversations on climate, with some large swathes of communities largely unaware (Nongqayi et al., 2022) of the scale of the challenges and with access to information being held by a suite of gatekeepers (e.g. Parsons and Traunter, 2019). For example, although young people have a right to access accurate information and participate in the development of their collective future (c.f. UN Convention of the Rights of a Child), young people who are aware of the climate crisis are still facing significant challenges to get their voices heard and valued, despite the rise of the #FridaysForFuture movement (Jones et al, 2021). Moreover, the broader societal perception of those young people that are engaged with addressing the climate crisis via activism, has been through a lens of being disruptive and troublesome (Fisher and Nasrin, 2021). A Global Action Plan survey conducted in 2020 explored >900 young people's engagement with social action, with key insights from the study identifying how the engagement of young people with climate change was dependent on other people around them and how much they "cared". This was found to translate into a belief they can make a difference if they also "cared", leading to further engagement and action based on shared compassionate values (SCV). Such collective belief and sense of belonging is vital in overcoming a climate value perception gap across society and fostering action on climate change, particularly across intergenerational relationships (Lawson et al., 2019).

There are a number of challenges in accessing the correct information and education concerning climate change. Resources and textbooks can quickly go out of date and even the best trained teacher can struggle to keep up with the fast-moving interdisciplinary fields which address climate change. However, through effective collaborations between research academics and teachers (Adamson et al, 2021) this gap can be closed and can also be very powerful when brought to the classroom (Monroe et al., 2019). It is well known that a scaffolded engagement in a learning environment can create personalised experiences for the learner which are key in changing behaviours and attitudes (Cordero et al., 2020). Van de Linden (2015), for example, has shown that addressing global issues such as climate change is best delivered through making the information local and contextualised rather than global and disconnected. This was shown to engage learners more effectively into the broader issue, leading to improved understanding and participation. Despite the importance of educating children and young people on the threats of climate change, engagement and enhanced understanding of these critical environmental issues have been shown to create eco-anxieties within young people (Halstead et al, 2021; Hickman 2021). However as illustrated by Halstead et al. (2021), self-belief and taking action with like-minded individuals can help address these feelings of anxiety and issues. Emotion and the climate crisis therefore are intrinsically linked (Halstead et al 2021; Meijnders et al., 2001; Nabi et al., 2018; Nairn, 2019; Ojala, 2018; Smith & Leiserowitz, 2014) and need to be incorporated into the methodologies of engagement with these potentially upsetting issues.

This paper reports on a project that sought to creatively embed climate education into a year eight (ages 12 to 13) secondary school class within a disadvantaged coastal community in East Yorkshire, UK, with the aim to build intergenerational dialogues concerning climate change and coastal erosion within the broader community. The East Yorkshire Holderness coast is one of the fastest eroding coastlines in the world. with average rates of retreat of >2 meters and local rates of up to 10 m a





year (Hobbs et al., 2020). The soft glacial till coastline means that erosion along this zone of coast has been happening for
centuries with many lost villages (Pye and Blott, 2015), but climate driven accelerations in rates are evident and set to further
accelerate non-linearly into the future due to sea-level rise and storminess (Kirby et al., 2021). As part of the work herein, the
aims were to understand what the local community thought about their changing coastline, what were the intergenerational
dialogues across the community in regard to coastal change, and did they recognise that climate change was already impacting
their town and that as a result they were effectively at the front line of climate change risk.

Climate change education, and in particular a climate literacy-based approach, seeks to enable students to become active
participants within the issue giving them the ability to identify, understand and explain information that is associated with
climate change. In turn, this ensures the students are best prepared for the challenges that they face into the future and having
the knowledge to enable them to consider and derive potential solutions (Lawson et al, 2018). Indeed, Miléř and Sládek (2011)
highlighted over 10 years ago a growing gap between what is known about climate change by the scientific community and
what is understood by the public. This gap has perhaps begun to close over recent years, but there remains significant need for
new ways to communicate and connect communities and the wider public to the climate crisis (Nisbet, 2009; Hügel and Davies,
2020). This gap has led many to advocate for a need to forward a climate literacy approach (Wu and Otsuka, 2021). Herein
we develop and detail an approach that sought to build climate-literacy within the school curriculum using a participatory
action research-based approach (Kennelly et al., 2023). We aimed to impart the basic knowledge of climate change and its
impacts which had a both a broad, global, scope but with a local focus and coloured by local and regional climate change
impacts. We additionally detail an exploration phase that was delivered through a suite of creative engagement opportunities
that were founded on these local impacts of climate change on the participants' communities, which was augmented by building
and facilitating intergenerational community dialogues. We also detail progress towards an aim that sought to foster positive
action and explore how the children and young people were supported in being creative in taking actions to communicate risk
and the impacts of climate change across their wider community. This final element took the form of a creative performance
and aligned short film which was presented to COP26. Herein we report and detail the approaches adopted within the
programme and critically evaluate the methodologies employed. We highlight the outcomes from the activities and critically
discuss the implications for climate change education and border geoscience communication within the context of a
disadvantaged, at risk, coastal community.

## 2. Methodology and Approach

### 2.1 Context, Positioning and COVID19 impacts

This project evolved from an ongoing relationship between project lead researcher Katie Parsons and a Withernsea High
School teacher, and co-author, Sarah Harris Smith. Ongoing work had sought to explore ways to encourage teachers to move





outside of the classroom and creatively use the outdoors in everyday teaching, looking specifically at the barriers that teachers face in doing this. One of the objectives of this wider project was to engage students with their own communities and wider landscapes they lived in, with the rapidly eroding Holderness coastline, being a key focus. The project aimed to understand children and young people's climate change knowledge and to understand the lived experiences of their community, and how, in turn, these experiences have impacted their lives. The work reported herein evolved from this framework.

Due to the impact of COVID19 limiting classroom access, we delivered online sessions each week for just under an hour via Microsoft Teams to two year-eight geography classes over a period of six weeks, with fifty-six students across the two classrooms. We utilised the interactive whiteboard functionality of Teams for the students to be able to see diagrams and interact via live visual displays. Due to covid regulations within the school at that time the young people had to face the front of the classroom and sit with a vacant space next to them. The limitations of the researcher not being there in person and

boundaries in place impacted the amount of group work that we would have liked to have undertaken. As a result, the discussions that took place were class wide and were supported physically within the classroom by the teacher.

Prior to the sessions taking place we asked all the students to complete two baseline questionnaires. One was led by their teacher on coastal erosion processes that they would be expected to know based on the geography curriculum and the other questionnaire was designed to understand the students' knowledge base regarding climate change, along with garnering an

understanding of the broader relationship each of them had with the outdoors and the environment. These same questionnaires were completed again at the end of the project in order to assess the learning outcomes from the sessions.

## 2.2 Participatory Action Research Sessions

The structured sessions used Participatory Action Research (PAR) approaches and was broadly based on Freire's theory of critical consciousness' (Figure 1; Jamal, 2017), which combine to consider how there is a need for '*reflection and action upon*

*the world in order to transform it*' (Freire, 1970: p51) and the degree to which individuals are able to "read" social conditions critically and in turn feel empowered to act to change those conditions (Godfrey and Grayman, 2014). Freire's praxis was articulated in his seminal work 'Pedagogy of the Oppressed' (1970: 65) where he adds:

> *"It is only when the oppressed find the oppressor out and become involved in the organized struggle for their liberation that they begin to believe in themselves. This discovery cannot be purely intellectual but must involve*

*action; nor can it be limited to mere activism, but must include serious reflection: only then will it be a praxis."*

It can be argued that children and young people can be considered as oppressed because of the climate and environmental injustices they face (Haywood, 2020). They have contributed little to the crises and also have the least opportunity to redress the issues through normal socio-political arenas. A greater critical consciousness within groups of children and young people has been related to improved mental health, better occupational outcomes and increased participation in more traditional types

of civic engagement (Heberle et al., 2020). This approach therefore provided a central tenet within the deployed methodology





in which we could allow the students to have an open dialogue to explore climate change and the environmental issues they face within the community to promote positive engagement.

Freire (1970: p52) notes that for praxis to be realised 'the oppressed must confront reality critically, simultaneously objectifying and acting upon that reality' and adds that 'critical and liberating dialogue, which presupposes action, must be carried on with the oppressed at whatever the stage of their struggle for liberation' (p.65). As such, herein we ensured that there is knowledge transfer and engagement as a first step and that the learning is scaffolded in order to support taking action. Discussions of climate change were not approached from the perspective of delivering scientific facts, but from the perspective of children's rights-based access to information (Articles 13 and 17, UN CRC).

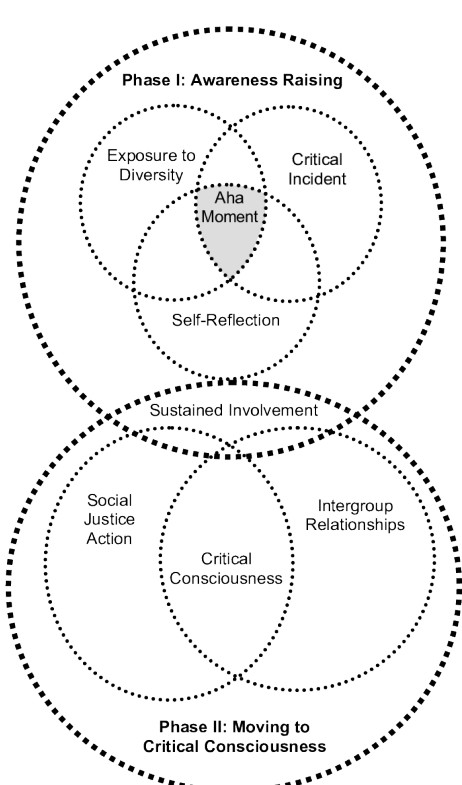

**Figure 1: Conceptual framework of critical consciousness**

Within the six sessions, the first session addressed the background interaction and introduced us to the class. We also explored the meaning of community and the young people's view of their place within this. The purpose of this exploration of place was central in grounding the young people to explore their sense of what place meant to them and incorporate their own lived experience within these first sessions. Through using this PAR method, we were able to show how: "*The lived experience of*



*place manifests itself in everyday activities, such as behavior, routine, social interactions, and cultural rituals*" Little (2020, p.26).

To facilitate this discussion, we undertook an interactive community mapping (e.g. Shkabatur, 2014) exercise and embarked on our own version of empathy mapping (e.g. Cairns et al., 2021) to elucidate their thoughts and feelings on the meaning of people and place. Using mapping methodologies with young people allowed them space and power to represent their own lives on their own terms, which has been shown to be important in engaging such participants (e.g. Brown et al., 2019). The session ended with a request that they make something creative that tells story about them.

In session two, we discussed the community maps they had completed from the previous session and asked them what they thought these maps showed us. We then compiled a word cloud based on the empathy mapping exercise and discussed the creative content for the words and the content. Moving on from this we looked at a range of purposely chosen photos and asked the students to tell a story based on their thought-provoking images. The second half of the session involved watching short news clip of an interview of a family impacted by coastal erosion and asked them to think of how they could represent that story in a photo. Finally, we left the session by giving them the start of a story and asked them to individually finish off the story in their own words before the next session.

Session three focussed on knowledge exchange and we started with explaining what climate change was, the physical processes and how it is impacting the planet. A "meet the expert" presentation comprised a 10-minute talk by an active researcher on climate and coastal change which detailed what was happening to the Holderness coastline and the likely future trajectories with projected sea-level rise. We ended the session by talking the class through the functionality of an ArcGIS Survey 123 mobile phone application we had set up for them to explore their local environment and record their findings in real time into the app. The app allowed them to undertake this activity without the pressure of working under teacher supervision or within peer groups, freeing their minds to experience their locale anew (Horton et al., 2014). The app allowed each participant to record their own narrative, with the possibility in the app to capture geo-located images or video and record evidence of environmental change in their community and describe how they were feeling at the time.

In Session four we shared the data from the ArcGIS Survey app (see Henning et al., 2023) they had used and talked through what they had found and discovered. The overall session aim was to enable the students to be in position to find and record stories from their local community. Within the session we therefore discussed storytelling as a research tool, what this was, and why it was so powerful to use. We also discussed gaining consent, ethics, and research interviewing skills. We additionally explored photography skills and how to capture the right photo and what you had to think of in photo composition. Finally, we set them the task of going out to find stories from their local community, family and friends and start to think of how they could best represent that story.

Session five was developed in partnership but was entirely delivered by the teachers as the young people needed to have space to focus on the development of their stories, with in person support. Within the session the participants looked at profiling





characters, using emotions to relay a story and exploring different ways you can creatively tell a story. Finally, in session six the young people shared their stories back with the entire group, with each participant taking turns to read out their work and tell us why they had chosen a particular story to tell.

Alongside the suite of workshops conducted within the secondary school, we advertised the project via the local newspaper
informing the wider community of the project aims and asking members of the public to send us in their stories, photos and lived experiences from the local region.  We were sent many creative representations of the communities' stories ranging from old photos to a poem, some amazing drone video footage of the region and active coastal erosion observations and even a recording of a folk group's musical performance, written about the troubles of living on the Holderness and the issues of coastal change.

After the workshops had taken place and with the help of the young people, we pulled together all of the stories that the communities had provided, alongside the young people's work from the school-based sessions. With the school participants, through co-creation sessions we began to interpret the information provided, before collating the stories. We collectively decided that compiling a film to share and exhibit the work would be an effective way to portray the discussions to a wider range of community members and broader audiences interested in the work. The film was shortlisted in the Climate Crisis
category of the Arts and Humanities Research Council's (AHRC) Research in Films Awards (RIFA) 2021 and was included in an AHRC led session at COP26 in Glasgow.

## 3. Results

The results of this project look chronologically at the methodology detailed above to understand the processes that the young people went through as they explored the impacts of coastal change and living in a changing climate. We explore the co-
creation of their journeys and provide observations and analysis on their engagement and how the young people's understanding grew and took different directions as the project evolved.

## 3.1 Climate change knowledge

Before the researchers met the young people, we wanted to find out what their knowledge base was concerning climate change. We explored what they knew, what climate change was, and if they able to contextualise it.  Additionally, we determined what
relationship they had with the outdoors, did they enjoy spending time in the natural environment, and was that bonded relationship already there.  The results of this exercise are presented in Figure 2 as word clouds for each of the questions asked. It is striking that the dominant response from the participants was uncertainty in the group, with "don't know" the most frequently selected response (36%). However, a number of the participants did identify temperature change, as well as melting ice and weather changes. In response to a question on what is causing climate change, the most frequent response was
"weather", followed by "people" and "don't know". The final question on who is responsible for climate change, the most




frequent response was "people", followed by "don't know" and "government" (Figure 2). A suite of misconceptions and a general lack of understanding was clearly apparent in the group responses. Similar outcomes from comparable surveys have been reported elsewhere; that despite the scientific consensus on the importance of responding to climate change, public understanding of climate change remain remarkably low. In the United States, only half of Americans view climate change as

a personal risk (Akerlof et al., 2010) and a more recent international study (Malla et al., 2022) highlighted that 38% of people surveyed still did not have an awareness of global warming. The outcome is thus a confused landscape with competing demands and public misconceptions (Eden, 1998) and an overall lack of engagement with the ongoing climate crisis. This is evidenced in the responses in Figure 2, with litter featuring prominently as a cause of climate change.

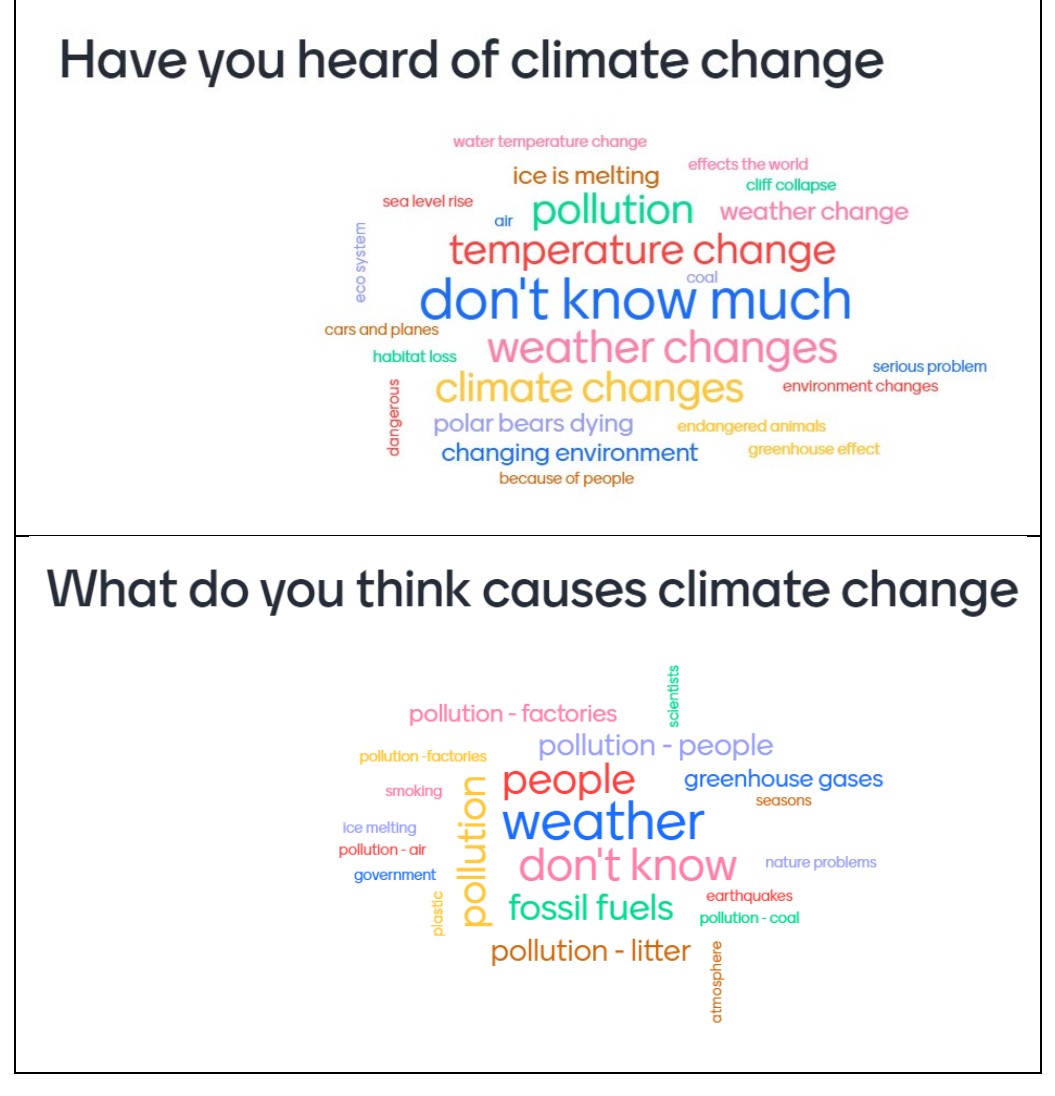





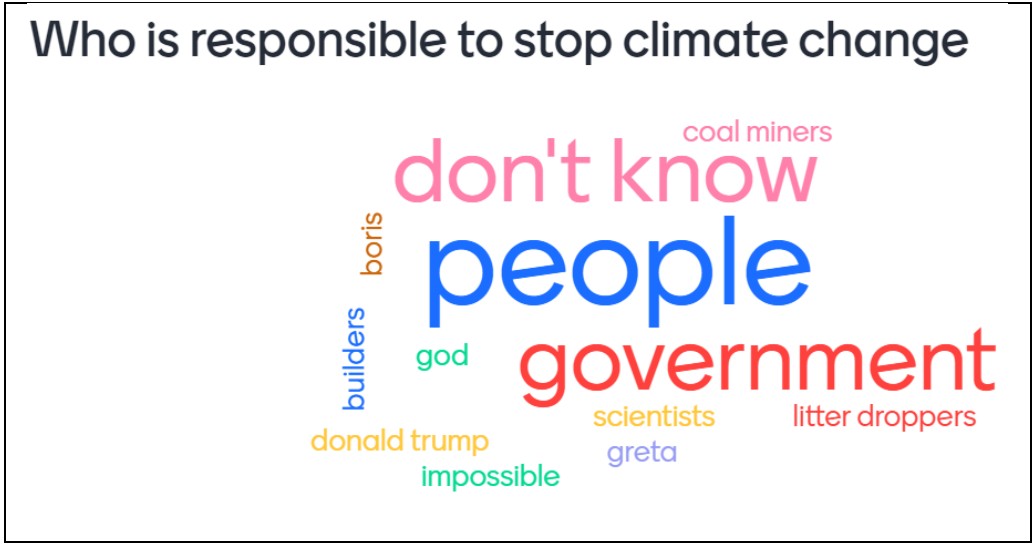


**Figure 2: Word clouds depicting young people's knowledge on climate change issues**

Such trends and confusion have been noted in similar investigations in the past (Truelove and Parks, 2012) and others have highlighted how a focus on tackling ocean plastic pollution in the public consciousness can provide a convenient truth that distracts us from the need for more radical changes to our behavioural, political and economic systems (Stafford and Jones, 260 2019) that addresses the climate crisis as well as the causes of plastic pollution, namely over-consumption. Most importantly, they did not view climate change as a risk to them and their community. Overall, the knowledge base of the participant group was relatively poor, with a passing understanding of climate change and its related challenges. These understandings were often distant, sometimes misconceived and with an overall limited engagement in the group with climate-related issues and sustainability more broadly.

**3.2 Community Mapping**

We asked the students to begin to map their place within their community, requesting they draw a map of Withernsea and what it meant to them. No two maps look the same, with each participant having different relationships and meanings with the places that surround them. Using their maps, we then asked the participants to mark areas on the maps using a variety of prompt questions such as identifying areas which young people go to, where do elders in the community frequent and where families 270 tend to go as a group. This task was designed to make them start to think about the different groups that live in their community and how they also interact with the spaces around them, additionally raising questions on whether any of these spaces were contested, what it feels like to be in these spaces and who is responsible for the different spaces they identify. We additionally explored these areas in groups to identify areas which the young people might class as dangerous or areas they do not feel safe in. A sub-set of the map images are represented below in the Figure 3.





The results of the community maps identify a range of perspectives, with all the maps covering different spaces and highlighting different relationships to place (Jagger. 2013; Fang et al., 2016). The maps exhibit different scales, with elements most important to them and their lived experiences being larger and amplified within the map. The participants were asked to identify on the maps how the different spaces were used, when they were used by different groups. This highlighted that the participants were aware of contested spaces and the use of different spaces by different groups at separate times of the day.

The maps were used as vehicle to also have the participants think about the utility of spaces and how these spaces were changing (e.g. best shown in Figure 3B - erosion, unused buildings, pollution; waste). The fact that this task was more difficult for the students revealed that environmental issues and features within their everyday spaces were in their hidden consciousness (Horton and Kraftl, 2017).

**Figure 3: Example community mapping outputs**




### 3.3 Empathy mapping

As highlighted above empathy mapping has traditionally been used in computer programming and marketing strategies to put the consumer at the heart of the decisions. These maps were produced by the participants in order and with a purpose of catalysing the young people to begin to explore how others in their community felt about their locale such that they (and myself as researcher) could start to understand the intergenerational and cross-sectional stories. Figure 4 shows examples of these outputs.

The empathy maps (Figure 4) revealed that the participants had similar views on the impacts of coastal change regarding power dynamics through their community. They mostly thought that both themselves as young people within the community, and older generations across the community, could do little to affect change. They identified that the elders within the town possibly felt "*scared and might feel angry*" and "*old people might feel sad because they can't do anything.*" The maps also showed that although they thought the businesspeople (Figure 4) (men in their maps) had most power to influence the future within the community; they also were viewed as having the ability to either develop "*plans to move away*" through to being "*worried and could lose their business.*" Their empathy maps concerning tourists revealed that although they believe that the tourists do identify the problems associated with coastal change and rapid coastal retreat, most would not be motivated to do anything about the changes given their temporary engagement with the area, summarised as "*I don't think they would do anything about it because they are tourists, and it is not their problem because they don't live there. They would probably take photos.*" One interesting outcome covered their empathy maps of adults in their community where one participant responded that an adult would say "*it's a normal thing*" but actually thinks "*it's getting worse.*", this could be an acknowledgement that children and young people often feel excluded and disempowered from denied to information concerning their lives. Overall, the responses for the groups across the participants' empathy maps tended to be short and also largely negative in tone, highlighting perhaps an empathic and assumed disconnect across groups and generations regarding climate change impacts, which has been identified elsewhere (Hayes et al., 2022).





**Figure 4: Example empathy maps**

### 3.4 ArcGis survey 123 – outdoor investigation

Using ArcGIS survey 123 a flexible spatial survey was designed to enable young people to collect their own data in real time. As part of the participants' homework, we wanted them to start to explore their own environment, outside, right on their doorstep and see if they could identify elements of their locale using the knowledge, they had learnt over the last few weeks within the workshop sessions with a view to create a story. The participants were then asked to describe the images and videos they had taken and identify the changes do they see and why are they think that these changes might be important.



Figures 5 A-D show an exemplar set of the data the participants collected. Figure 5A was described by the participant as an image showing "People moving the sea defences in front of the cliffs" and was highlighted as important as: "these changes are going to slow down coastal erosion." In response to the same prompts the owner of Figure 5B responded, "a field" and that

this was important as it has: "shrunk" and there would be: "not be enough room to harvest [in the future]". The proponent of Figure 5C took this image as it depicts "the rocks that the sea is bringing in and is tearing the cliff edges off" and that there was now a: "a lot less sand but a lot more rocks", which might be important because this: "could possibly slow the sea down?" Finally, in Figure 5D the participant noted that the image depicted "the cliffs and that it has been eroding quite a lot and there is a lot of parts of the cliff at the bottom", they noted in discussion that "it has changed because it wasn't as eroded as before

and more parts of the cliffs have come off the bottom", they noted that these changes were important because: "it shows how bad coastal erosion is and that we need defences in and to show and teach people about coastal erosion."

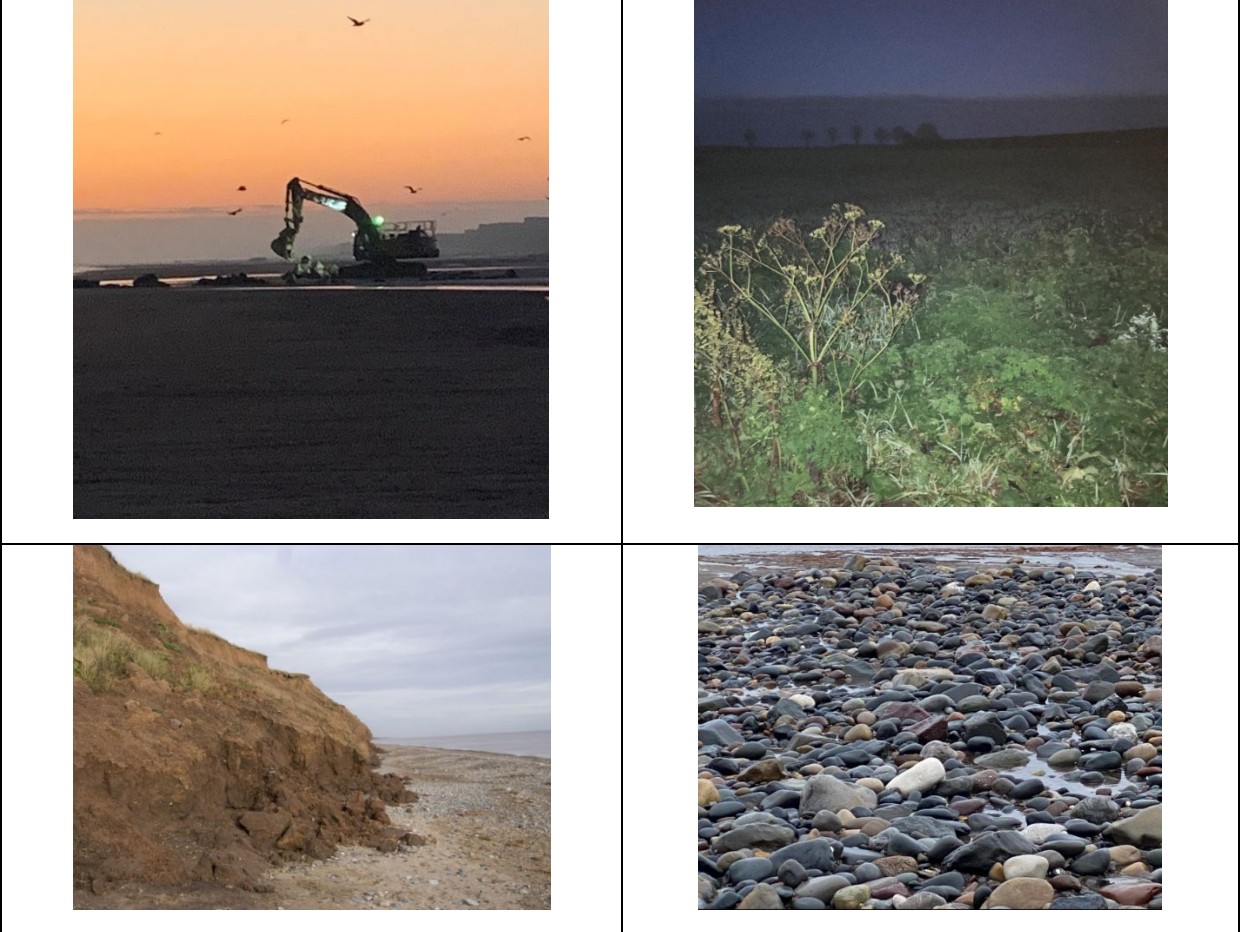

**Figure 5: Photos from the participants' ArcGIS 123 Survey (clockwise from top left A-D)**



In discussion their data collection and observations, the teacher highlighted how the group were:

> *"obsessed with the fact that the sea is brown... when you look at photos of seaside's, you don't see the brown sea and they are absolutely obsessed with why the sea is brown... And I said that's got to do with coastal erosion and they made that an assimilation that the boulder clay, obviously as it was eroded from the cliffs it infects the sea and they felt that as infection. I felt they were using the words infection or pollution, it should have been pollution really, but*

*they were saying that it was infected and that really gets to them. They feel that people... well that contributes to the image of Withernsea.  Well, that's where the seas brown, why would you want to go there? Its infected, its that whole disregard for Withernsea as a place as well.  Cause they know that a lot of people don't hold it in high esteem, and they think that the brown sea contributes to that and obviously, they don't see that the sea is creating that."*

The important outcomes of this observational exercise were the clear evidence of a widening of their local environmental

knowledge and engagement with the impacts of coastal change in their community. Within the new praxis journey the participants were on a looping journey between exploration and knowledge, reflecting what they had learnt through playful and creative encounters with their environment.

**3.5 Co-created Thematic Film**

The final output from the participants was a film (Figure 6) that was created as a way to show and exhibit what they had found

and tell their collective story about their locale. The original plans were that the students were to hold a final exhibition in a local space which showed their work through photos and poetry. However, due to the COVID-19 lockdowns and public spaces still being closed to large gatherings, we decided that we wanted to put our exhibition online and make it digital. At first, we co-developed plans for digital exhibition spaces, but then the idea of digital montage started to form. What began as a montage became a film once we started to pull all the data together.

There was a real sense of privilege in that the young people were allowing us to use their voices, photos, and data to showcase the place they called home. When the first cut was ready, this was sent to the school for feedback and if they felt it represented what they wanted to say. The video depicted what was important to them and told of the intergenerational connectivity fostered by the project. Most importantly the film showed emotion and produced an emotional reaction within the participants. Previous research has shown how empathy can be a key motivator for climate action (Jones et al. 2021; Hayes et al., 2022), with one

way of promoting empathy being the communication of a localised, place-based, people and community-focused stories of climate change impacts (Swim and Bloodheart, 2015). The film showcases these approaches and outcomes, which will be further discussed below.



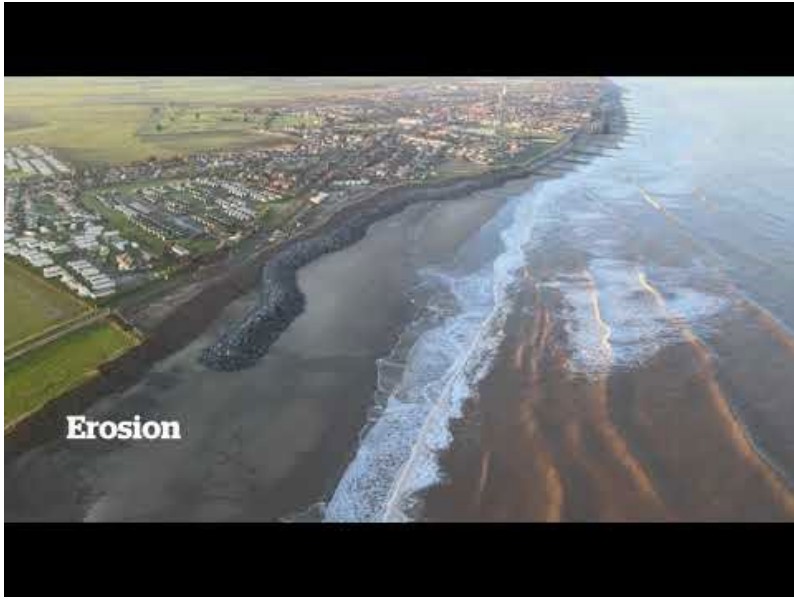

**Figure 6: INSECURE Project summary film**

## 4. Discussion

The results record the journey the student participants went on as the programme of workshops progressed and they increased their knowledge of climate change and engaged with wider environmental issues in their locale, along with their own place(s) (with)in their community. A set of themes emerged from the research that have a suite of implications for engaging at risk

coastal communities. This included analysis of the need for a broader-based climate education that has local context embedded and that this needs building to build and engender an understanding across intergenerational and community-based dialogues.

### 4.1 A new Climate Praxis Model

The work through the series of workshops and engagements has effectively developed and deployed a methodology that can be best represented as a *new climate praxis model* (Freire, 1970) based on knowledge (gain), exploration (play) and action

(Figure 7), which combined connectivity to an environmental issue (coastal change) through to adaptation. The work was undertaken in partnership as educators and researchers, and we ensured that this was both age and socially/culturally-appropriate, building a 'critical dialogue' and recognising people's lived experiences. This asset is central to Freire's notion of praxis and forms an important shaping influence on the model. Reflecting on how this progressed through the methodology and the results, it is evident that this advanced through a double looping learning process and journey (Trajber et al., 2019),

where a reflective approach within the cohort enabled participants to gain knowledge and understanding, explore and widen their perspectives through to beginning to take action on coastal change within their broader community. Additionally, the





interfaces between the principal components include looping and reflexivity, essentially involving each of the participants transiting their own "wave of change" (Jones et al. 2021). There is evidence in their journey as individuals and a collective of macro-level looping, where additional reflection leads to deeper knowledge, enhanced engagement and results in amplified
action (Figure 7).

This new model and the participants' journey in the study therefore very much has its theoretical foundations in Freire's (1970) seminal work outlined in the 'Pedagogy of the Oppressed.' This work has had a profound influence on educational practice around the world. Freire argues that traditional education is inherently oppressive, because it treats students as empty vessels to be filled with knowledge by the teacher. In his approach Freire proposes "*problem-posing education*," which treats students
as active participants in the learning process. Freire's ideas have been praised for their insights into the nature of oppression and the power of education to liberate people. Although these ideas have also been criticised for being too idealistic and for ignoring the realities of power in the classroom, one of the key strengths of Freire's work is his focus on the importance of dialogue in education. he argues that students and teachers must engage in dialogue to create an effective learning environment, allowing students to share their experiences and perspectives, developing critical thinking skills. Another strength of Freire's
work is his emphasis on the importance of praxis, upon which the new model developed has its foundation. Praxis is the coupling of action and reflection, which is argued by many as essential for social action and transformation (Bang and Vossoughi, 2016). Freire (1970) argues that education must be based on praxis, so that students can learn from their experiences and take action to change the world (Clark, 1993).

In the context of the climate crisis, it has been argued that children and young people can be considered as oppressed because
of the climate and environmental injustices they face (Verlie and Flynn, 2022; see UNCRC General Comment 26[1]). This is particularly the case here where the participants live on one of the fastest eroding coastlines in the world and were largely unaware, disengaged, and had little understanding of the possible impacts of climate change on their town into the future and within their lifetimes. A greater critical consciousness within groups of children and young people has been related to improved mental health, better occupational outcomes and increased participation in civic engagement. Freire (1970: p52) notes that for
praxis to be realised *the oppressed must confront reality critically, simultaneously objectifying and acting upon that reality*.' Through the workshops the participants were introduced to a suite of new knowledge through a PAR based approach. This engagement was a necessary first step, scaffolding the learning and supporting the exploration of coastal change and the impacts of climate change on their town and community into the future. Thus, one could argue that children and young people are not only oppressed by the ways in which they are presently educated about climate change, but are also oppressed by the
ways in which climate change will impact their lives in the future and how their voice and action is not presently impacting policy in either realm.

---

[1] https://www.ohchr.org/en/documents/general-comments-and-recommendations/crccgc26-general-comment-no-26-2023-childrens-rights



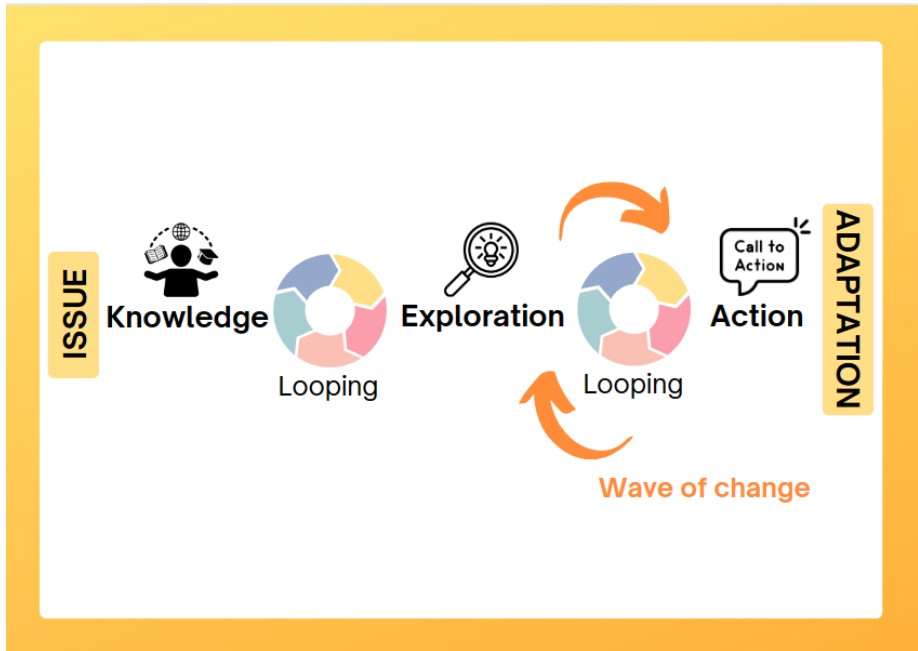

**Figure 7: A new climate praxis model of knowledge, exploration (play) and action, incorporating Jones et al. (2021)**
410                                          **Wave of Change**.

Perhaps most critically, the new climate praxis model developed within this paper includes emotional waves of change (Jones et al. 2021; Figure 7) within the looping interfaces between "knowledge and exploration" and "exploration and action", where individuals process the implications of change, and the disruption likely to their 'normal life' (McAdam 2017). Based on a Kübler-Ross (1969) Change Curve, the wave of change (Jones at el., 2021) identified stages of grief and loss that individuals

experience on understanding the scale of the climate crisis, moving through Denial, Anger, Bargaining, Depression and Acceptance.

Many of these traits are evidenced in the creative engagements of the participants, which had negative connotations such as the largely negative empathy maps and ArcGIS app exploration journals and photos, perhaps highlighting that the cohort were beginning to understand, via looping, the implications of coastal change on their communities. The lack of *apparent* personal

and direct experience of the impacts of global warming evident in the early stages of the programme of workshops has been argued to foster and shape these views. The negative impacts of climate change are often viewed as only possible in a distant future that affect people, species, and places far away. This results in a psychological distance that forms a major barrier to climate action (Spence et al., 2012). Although, more recent research has cautioned against over-estimating the impacts of a psychological distance (van Valkengoed et al., 2023), the impact of human exceptionalism (Betz and Coley, 2022) playing a

role cannot be ignored. In their study Betz and Coley (2022) note that although their participants correctly think that humans





uniquely contribute at scale to climate change they conversely, and incorrectly, believe that humans will uniquely be protected from the impacts of global climate change. This human exceptionalist thinking is also noted to likely reduce engagement and action on climate change.

## 4.2 Understanding, Empathy and the Local

Through the workshops the participants began to understand the possible impacts on them and their communities, as evidenced in the responses to the ArcGIS surveys, where photos identifying specific impacts were collected with clear rationales. Jones at al. (2021) propose that acceptance stage of their wave of change can be viewed as a rebuilding stage and is significant in understanding the emotional journeys in climate activism and social action. McAdam (2017: pp.194) highlights that in order to mobilise people into a movement, '…*at a minimum, people need to feel both aggrieved about (or threatened by) some*

*aspect of their lives and optimistic that by acting collectively they can begin to redress the problem…*'. In trying to understand this lack of mobilisation on climate, despite the existential threat, McAdams (2017) argues that people are not as concerned about climate change as they should be because it can be seen as a distant issue (psychological distance) and not experienced sufficiently by the individual, alongside the scale of the challenge being seen as overwhelming and discouraging. There is a need to find ways to make climate change more relevant and personal to people, simplify the issue and engender emotion.

Again, these align with the journey of the participants through the workshops, with many disengaged from understanding the impacts of climate change and coastal change in the early sessions through to being motivated to take action towards the end of the programme, through to becoming activists and promoters of their film telling their communities' story to a wider audience.

Previous research has shown that empathy is a key motivator for climate action (Jones et al. 2021; Hayes et al., 2022). One
way to promote empathy is to communicate localised, human-focused stories of climate change impacts (Swim and Bloodheart, 2015). However, climate change communication and education has typically applied what is termed an information-deficit model (Suldovsky, 2017), which many have argued rarely evokes the deep emotional responses required to lead to action (Bloomfield and Manktelow, 2021). This information-deficit was clearly evident in aspects of the teaching group experiences, with one teacher commenting "*Community mapping, they don't see that as Geography… when they come in the classroom*

*they expect Geography.*" However, the participants engaged incredibly well with the materials where the impacts of climate driven coastal change were made local to them and how this coastal change would impact their locale. Many students commented (via the teacher): "*The students enjoyed it, I enjoyed it*" and "*He has been very engaged when he usually isn't.*" "*I have a student that is electively mute, but since taking part in this project she has chosen to speak, she even spoke to the BBC interviewer when they came to school. The whole class has been so engaged... students who do not usually do well in*

*Geography are now wanting to take it as a subject. I will definitely be looking at putting creativity into our modules as the engagement has been fab*" This deliberative, localised, place-based approach, reduced the opportunity for amplification of the spatial and temporal distancing (McAdam, 2017) that was evidenced in the questionnaire responses from workshop 1.



Wardekker and Lorenz (2019) have also previously noted how the IPCC's framing of the climate crisis is often problem-focused rather than solution orientated. De Meyer et al. (2021) have similarly advocated for a more solution-orientated focus, specifically linking this to storytelling and the importance of stories of hope, that is, framing climate change within positive, more hopeful examples of recovery and adaptation. As described in Hayes et al. (2022) '*Hope is key to sustaining collective resistance*' and '*does not rely on optimistic emotions, but allows reimaginations of a future*', which can encourage both individual and community action (Halstead et al., 2021; Jones et al., 2021).

## 4.3 Action and outcomes

Storytelling has long been held as a catalyst for change (Neile, 2009; Wånggren, 2016), leading Zingaro (2009) to note that storytelling's substantial transformative potential, can in itself, be a significant form of taking social action. The participants in this study certainly felt that the film told their story and that this was a form of action from them. The young people participants have emotionally connected to the issues that were important for their locale and taking ownership of the ideas of taking action in the here and now, instead of leaving it to others, elsewhere, in the future (Gifford, 2011). Hicks (2018) discusses how a sense of agency can be nurtured through learning, sharing and acting in partnership with others and Hayes et al. (2022) have outlined how empathy and hope can cultivate spaces for intergenerational dialogues. This shows that a community is more powerful when working together, with dialogues developing across communities that remove apparent barriers and fostering action. In turn the outcomes highlight how this can support young people in generating the hope that allows them to take ownership over their stories and a reimagining their future(s).

There is a set of important outcomes and learning that can be derived from the work. The model and approach to engagement has had a significant impact on all participants, including the researchers and teaching team. The exposure of the co-created film at COP26 and participation at the AHRC award ceremony has been an immense source of pride across the group and wider community[2]. Perhaps the most important outcome has been the way the participants responded to the sessions and the comments of the teachers concerning the impact of the engagement with the class:

> "*The students enjoyed it, I enjoyed it*" and "*...For me in the classroom what really, really surprised me ..and what I absolutely loved ..and gives me this little glow, is the way that students really expressed themselves... I saw a different side to some of those students...we don't tend to... they don't get that exploration, you don't get as much play time, .... But a couple of students really thrived in that free rein in how to present their work. [Boy7] really expressed himself and it turns out he's fantastic at English and it was seeing, I suppose marrying Geography with literacy in the curriculum...that creativity is something that I am going to explore more.*"

---

[2] https://www.hulldailymail.co.uk/news/hull-east-yorkshire-news/cop26-film-project-flying-flag-6149257





Building on how we understand emotions as evaluative feelings which are coupled to people and their environment in meaningful ways, increasing attention is being paid to the use of emotions and creativity in educational practice. This is particularly the case concerning communication about the climate crisis and other sustainability challenges. Dunlop and
Rushton (2022) draw on data from teachers, teacher educators, and young people to describe how emotions can be utilised to enhance engagement in educational settings. Their results highlight how emotionally responsive pedagogies are needed to identify responsibilities, build resilience and shape imaginations of the future. A more enabling educational policy environment is needed in the UK such that teachers and educators are able to adopt approaches similar to those pioneered here and thus empower them and the young people in their classroom to take action relating to climate change impacts.

## 495 5. Conclusions

The project reported on herein demonstrates the transformative potential of creative, participatory approaches to climate change education in disadvantaged coastal communities. By engaging young people through intergenerational dialogues, place-based learning, using creative methodologies, the work has effectively bridged the gap between scientific knowledge and lived experience. This approach not only deepened participants' understanding of climate change and coastal erosion but
also empowered them to take meaningful action within their communities, challenging traditional information-deficit models of education.

At the heart of the project lies the development of a new Climate Praxis Model (Figure 7), rooted in Freire's (1970) principles of critical consciousness. This model integrates knowledge acquisition, exploration through play, and action, supported by emotional engagement and reflexivity. Through interactive workshops, community mapping, and the use of digital tools like
ArcGIS Survey123, participants were able to connect their personal experiences to broader climate narratives. The culmination of these efforts—the co-created film—provided a platform for young people to share their stories with wider audiences, fostering a sense of agency and collective identity. The research uncovered a set of themes that emphasised the necessity for a comprehensive, locally grounded climate change education and the importance of fostering intergenerational and community-based dialogues within this framework. The approach, looping through knowledge-exploration-action, evidently empowered
the students to critically engage with the pressing issue of climate change, particularly pertinent to their coastal town facing rapid erosion. The participants progressed through the model and a double-looping learning process, reflecting on their experiences and taking meaningful action within their community through a storytelling approach. This emotional nature of their story reflected in the participants' creative works, which although conveyed a sense of negativity, signalled an emerging understanding of climate change's implications and how they could take ownership of their story.

The outcomes reveal how localized, participatory climate education can address psychological distance, amplify empathy, and inspire hope, creating pathways for intergenerational dialogue and community action. The initial lack of personal experience with climate and environmental change impacts was compounded by a clear psychological distance between the causes and




effects on them and their locale. This presented initial barriers to engagement and climate action. However, as the programme and praxis model unfolded, the participants began to recognise the potential consequences for them and their community, spurring them into observation and action. This transition was rooted in the clear importance of making climate change personal and locally relevant to them and their communities, with empathy key to this engagement. Empathy was a central theme of the creative outputs and was weaved throughout the film, that centred on localised, human-focused stories of climate change impacts. This was witnessed to foster a sense of agency, intergenerational dialogue, and collective action. The participants took ownership of their stories and began to reimagine their futures, embodying the power of community collaboration and hope. Moreover, the positive feedback from teachers and the enthusiastic engagement of students highlights the importance of integrating creativity and emotional responsiveness into educational practices.

This work therefore underscores the need for educational policies that enable innovative, place-based approaches to climate communication. The success of the INSECURE project suggests that similar models could be replicated in other at-risk communities to foster resilience and agency in the face of climate change, using the climate praxis model as a framework or toolkit. By empowering young people to act as advocates and storytellers, this project has demonstrated how education can play a pivotal role in preparing communities for the future impacts of environmental change.

Participants shifted from initial disengagement and misconceptions to an empowered understanding of their role in addressing climate challenges.

In closing, this work has adopted Freire's pedagogical principles and integrated emotional intelligence to engage participants in understanding and addressing climate change within their at-risk coastal community. It has showcased the potential for transformative education to empower individuals and communities to take meaningful action in the face of pressing global challenges. The impact of this programme extended beyond the young people participants, leaving a lasting impression on the teaching team and the broader community of Withernsea. Ultimately my aspiration is that the legacy of the project serves as a beacon of hope, illustrating the potential of education to shape imaginations of a more sustainable and resilient future, where empathy, creativity, and collective action prevail.

**Author contribution**

KJP led the project, driving all key aspects including conceptualisation, methodological approach, workshop delivery, analysis, project administration, creative outputs, and also led writing both the original draft and subsequent revisions. FH supported the development of the methodological approach, workshop delivery and data collection, as well as supporting the curation of the creative outputs. LJ has been a champion for the project, enabling funding acquisition, providing advice and subsequently reviewed the manuscript. SHS facilitated the workshops, scaffolded the learning with the participants, and contributed to the method development.

**Competing interests**



The authors declare that we have no conflict of interest.


**Ethical Statement**

The research project was conducted in full compliance with ethical standards, and was approved by the University of Hull Ethics Committee. All research activities adhered to the university's ethical guidelines and relevant national and international standards for research involving human participants. Approval was obtained prior to commencing the study, with careful
consideration given to participant informed consent, confidentiality, and data protection.

**Acknowledgements**

We would like to extend our heartfelt thanks to the pupils of Withernsea High School for their invaluable participation and enthusiasm throughout this project. Your curiosity, creativity, and dedication to exploring environmental issues and taking action have been truly inspiring. This work would not have been possible without your contributions, and we are deeply
grateful for your insights and efforts. Thank you for being active agents of change and helping to shape a brighter future for our environment. We made use of AI in proof-reading the final version of the manuscript.

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
