# Peer review of "Crumbling cliffs and intergenerational cohesivity: A new climate praxis model for engaged community action on accelerated coastal change"

_EGUsphere, 2024_

## Author Comment (AC1)

**Reviewer 1:**

- I appreciated reading this paper, which tells a compelling and timely story about an action research project engaging disadvantaged young people in a coastal UK community. The authors have clearly undertaken a thoughtful, locally rooted project that addresses the urgent need for more inclusive forms of climate communication and education. The narrative is engaging, and the topic is relevant to diverse readerships. However, I believe that in its current form, the paper does not yet meet the standards of scientific rigour, theoretical grounding, and methodological clarity expected by a peer-reviewed journal. Major revisions are required for it to constitute a publishable scientific contribution. Below, I detail both key concerns and recommendations, structured to reflect the journal's evaluation framework.

>> We thank the reviewer for such a thoughtful and supportive critique of our work noting the timeliness, thoughtfulness and compelling place-based elements of our project. We thank the reviewer for the supportive comments on including highlighting areas for suggested improvements as well as helpful recommendations. We sought to tell the story of the work – given the Journal's ethos - but have added in a little more structure to the narrative in response to the reviewers suggestions. We respond to the specific suggestions inline below, noting the changes made to the manuscript.

**Scientific Significance**: The manuscript presents a meaningful case study and offers a potentially valuable contribution in the form of the "new climate praxis model." However, the theoretical and conceptual development of this model is not adequately established; the paper lacks clarity on how it advances the existing state of knowledge on the topic, particularly within the domains of climate literacy, participatory education, or community-based climate action, and how the model builds upon such literature. While the project is potentially original in its practice, the paper currently reads more as a descriptive – and engaging - account of an experience than a critical analysis of a topic, offering new conceptual tools, methods, or generalisable insights for future research and practice on the topic. Also, terms like "intergenerational dialogues," "creative methodologies," and "transformative potential" are used loosely and need clearer definitions, theoretical anchoring, and evidentiary support.

>> We have added an additional paragraph to a now extended introduction to capture the theoretical and conceptual development of the new praxis model and how this links to an extends elements of climate literacy, participatory education, and community-based climate action. This section reads: *"The work reported herein also reports on the outcomes from a methodological development perspective where a new climate praxis education model evolves and is refined from the engagements. The new model builds directly on established traditions in participatory action research (PAR; Cornish et al., 2023) and Freirean (Freire, 1970) critical pedagogy, advancing them in the specific context of climate literacy and community-based climate action with marginalised youth. The model is grounded in a critical engagement with theories of knowledge co-production, knowledge democracy (Duncan-Andrade and Morrell, 2008; Chapman, 2019; Stern, 2019), and communities of practice (Lave & Wenger, 1991), contributing conceptual advances through integrating rights-based climate education with storytelling as a method of participatory inquiry and intergenerational dialogues. Similarly, the creative methodologies serve as tools of knowledge mobilisation and repositions youth as legitimate knowledge producers and policy actors, thus offering a replicable approach to community-led*

*climate resilience that has already begun influencing local practices and responses. In this way, the model offers both theoretical and practical advancements in participatory climate education, positioning it as a new conceptual tool for future research and action. Herein we report and detail the approaches adopted within the programme and critically evaluate the methodologies employed. We highlight the outcomes from the activities and critically discuss the implications for climate change education, using the climate praxis model as a template for border geoscience communication within the context of a disadvantaged, at risk, coastal community."*

**Scientific Quality**

Several areas of the paper would benefit from improved methodological transparency and conceptual clarity:

- **Literature/theory integration**: The current literature is concentrated in the introduction and lacks a dedicated section clarifying key debates in climate education, youth engagement, or critical pedagogies. This weakens the paper's ability to make a clear and evidenced claim about its knowledge contribution. A focused literature review and stronger conceptual grounding – linking contemporary climate education debates with Freire's critical pedagogies theory - are urgently needed.

>> The extended introduction above also directly addresses this point linking Freire's critical pedagogies theory to contemporary climate education.

- **Conceptual clarity**: Terms such as "intergenerational dialogue" are not clearly justified or demonstrated in the empirical material. Since the project centres on year-eight students, it is not apparent in what way intergenerational exchange was achieved. Was the creative material from the community integrated in the sessions? Did the youth engage directly with elders at some point? These elements need to be clarified.

>> A sentence has been added to clarify this element of the work. We thank the reviewer for highlighting this as we note we implied rather than explained this. The sentence reads: Line 91: *"build intergenerational dialogues (from children to senior members of the community) concerning climate change"* And Line 225: *"Finally, we set them the task of going out to find stories from their local community, family and friends and start to think of how they could best represent that story and build intergenerational dialogues into the data collection."*

- **Methodological rigour**: The methods section does offer helpful detail on the sessions and activities undertaken with students, but it lacks critical information on:
    - **Sampling**: Why this age group? How were students selected? Were there any limitations emerging from this sample?
    - **Data sources**: What exactly is being analysed in the paper? Is this an analysis of the methods, the participants' outputs, or the overall process/methodology?
    - **Analytical strategy**: Are statements (e.g., "students did not perceive climate change as a risk to their community") based solely on outputs such as word clouds and maps? Were transcripts, field notes, or recordings used to support

interpretation? Clarification is needed on how claims were derived and what data supported them.

>> We have added in these elements of the methodology, which were not included to save space. We have also clarified the data sources as well as the analytical strategies employed. This section now reads: Line 140: *"The work reported herein evolved from this framework and the recruitment of the class was through the existing relationship with the school."* And Line 145: *"Session transcripts were recorded alongside session notes for onward analysis and to support later interpretations."*

- **Use of references**: Some citations are used imprecisely or overly broadly (e.g., at some point Freire is cited as if he created the "new climate praxis model," rather than the authors building upon his ideas). Greater care is needed in situating this work within—not merely citing—the theoretical literature.

>> We found the Freire reference and we have altered the text to cover this point but could not find other examples suggested by the reviewer. Nonetheless we have checked through the work and tested the assumed links to the theoretical literature. Moreover, in response to RC2, we have added in additional paragraphs to the discussion to link more fully back to the theoretical literature. Line 385: *"The work through the series of workshops and engagements has effectively developed and deployed a methodology that can be best represented as a new climate praxis model based on Freirean theory (Freire, 1970). This model builds a knowledge (gain), exploration (play) and action (Figure 7) framework, which combined connectivity to an environmental issue (coastal change) through to adaptation."*

- **Presentation Quality:** The manuscript is engaging but needs clearer structure and focus to guide the reader through the research story:

>> We thank the reviewer for these kind words of encouragement. We sought to tell the story of the work – given the Journal's ethos, but have added in a little more structure to the narrative in response to the reviewers suggestions. For example we have added a new section 2.3 on the workshops and the PAR theory – again to help signpost. This now reads: **"2.3 PAR Sessions -** Six PAR sessions were run, each with a specific focus. The first session addressed the background interaction and introduced us to the class. We also explored the meaning of community and the young people's view of their place within this. The purpose of this exploration of place was central in grounding the young people to explore their sense of what place meant to them and incorporate their own lived experience...."

- **Introduction**: Overly long and lacks a clear funnel from problem → gap → contribution → research aim → structure of the paper.

>> The introduction is longer as a result of the changes made, but we have added in sub-sections loosely following that suggested. Which we think addresses this need. This includes a new section on Climate Praxis.

- **Findings**: Results are described chronologically and narratively, which is compelling, but they need to be more analytically unpacked. For instance, the claim that students developed a deeper understanding of climate risks should be substantiated with clear evidence (e.g., comparison between baseline and post-questionnaires, direct quotes, etc.).

>> We have added in this detail as requested into the results section, including the additional quotes to justify the findings. See lines 260 onwards. Additionally the word clouds are direct quotes from the questionnaire and this has been made clear in the Figure title.

- **Language and terminology**: The paper uses accessible language, but at times imprecise or vague terms (e.g., "creative," "transformative") are left undefined.

>> We are not clear that these terms are poorly defined. We have chosen to retain their use as is in the paper to ensure that it continues to use accessible language, following the ethos of the journal.

**Recommendations for Revision**

To strengthen the manuscript, I suggest the following:

**Rework the introduction** to provide a clear and structured argument: define the problem, establish the knowledge gap, state the contribution, describe the methods briefly, and outline the structure of the paper.

>> The introduction is reworked, as suggested, with sub-sections included as outlined above.

**Add a dedicated literature/conceptual section**:

- Map key debates in climate education, especially with disadvantaged youth.

>> This is added into the extended introduction rather than a new conceptual section suggested, but as a sub-section at the end of the introduction.

- Understand the key concepts that the paper builds upon and clarify what they mean: "intergenerational dialogue," "creative climate engagement," and "transformative education," using scholarly sources. I personally do not think this paper is about "intergenerational dialogues", but I could be wrong. A clearer explanation would help.

>> We have tightened up the use of terms through qualification and the addition of citations to guide the theoretical interests, including in the new section in the introduction.

- Discuss the foundations of the "climate literacy-based approach" and how your model builds upon (or diverges from) this.

>> This is added in the discussion section, with additional links back to Freire and the theoretical constructs of the work. This section (Line 395) now reads: *". Reflecting on how this progressed through the methodology and the results, it is evident that this advanced through a double looping learning process and journey (Trajber et al., 2019), where a reflective approach within the cohort enabled participants to gain knowledge and understanding, explore and widen their perspectives through to beginning to act on coastal change within their broader community. In effect the work extends a climate literacy-based approach that seeks to enable students to become active participants and ensuring they are best prepared for the challenges that they face into the future with the knowledge to enable them to consider and derive potential solutions (Lawson et al, 2018; Hügel and Davies, 2020). Additionally, the interfaces between the principal components include looping and reflexivity, essentially involving each of the participants transiting their own "wave of change" (Jones et al. 2021). There is evidence in their journey as individuals and a collective of macro-level looping, where additional reflection leads to deeper knowledge, enhanced engagement and results in amplified action (Figure 7). "*

---

## Author Comment (AC2)

**Reviewer 2:**

This paper presents an inspiring and much-needed intervention into the world of climate education. The project's emphasis on creativity, place-based learning, and emotional engagement is both powerful and timely—especially given the urgent need to include younger voices and coastal communities in conversations about climate change. That said, while the initiative is rich in vision and community engagement, the paper itself reads more like a project report than a tightly argued academic article. Below are some thoughts on how it might be strengthened for publication.

>> We again thank the reviewer for these very warm and supportive words concerning out paper as well as the time and consideration to how the manuscript could be improved upon.

1. The paper would benefit from more clarity around why Year 8 students (~12–13 years) were selected. Was there something about their developmental stage that made them ideal for this kind of emotional and creative engagement? Offering a short explanation here could help readers understand the pedagogical rationale more clearly.

>> We have added in a sentence to address this point on the age range selection. It was entirely serendipitous in the work built on existing relationships. This is now made clear: Line 140: "*This project evolved from an ongoing relationship between project lead researcher Katie Parsons and a Withernsea High School teacher, and co-author, Sarah Harris Smith. Ongoing work had sought to explore ways to encourage teachers to move outside of the classroom and creatively use the outdoors in everyday teaching, looking specifically at the barriers that teachers face in doing this. One of the objectives of this wider project was to engage students with their own communities and wider landscapes they lived in, with the rapidly eroding Holderness coastline, being a key focus. The project aimed to understand children and young people's climate change knowledge and to understand the lived experiences of their community, and how, in turn, these experiences have impacted their lives. The work reported herein evolved from this framework and the recruitment of the class was through the existing relationship with the school.*"

In terms of data and analysis, the authors gathered an impressive range of materials—empathy maps, stories, community maps, and more. However, how these were analysed is a bit unclear. There's very little mention of how themes were drawn out or if any coding frameworks were used. It might help to walk readers through the process a bit more—what was looked for, how interpretations were made, and what might have been left out. The pre- and post-questionnaires mentioned early on sound like they could offer some incredibly valuable insights, but they're not really brought back into the paper later.

>> Additional methodological detail has been added. This was brief in order to ensure the manuscript was concise, but in taking a steer from R2, this additional contextual methodology has now been added. We have added this text to the start of the results section: "*The results of this project look chronologically at the methodology and sessions detailed above to understand the processes that the young people went through as they explored the impacts of coastal change and living in a changing climate. We explore the co-creation of their journeys and provide our observations, based on transcripts of the sessions, as well as our observations on their engagement and how the young people's understanding grew and took different directions as the project evolved.*" And Line 380: "*. A set of themes emerged from our coding of the*

*observations of the sessions and analysis of the creative materials produced, that have a suite of implications for engaging at risk coastal communities."*

2. There are some lovely themes in the paper—like intergenerational learning, empathy, and localised environmental awareness—but they don't feel fully developed or tied back to the conceptual framework. Since the paper gestures to Freire and critical pedagogy, it would be powerful to revisit those ideas when analysing the data: How did students engage in dialogue? What kind of transformation, if any, was visible? Some of the claims— for example, that students "took action" or developed empathy—are really compelling but feel a bit anecdotal. Could these be grounded more in the data? A quote here and there is useful, but a bit more structured evidence would go a long way in making those claims more convincing.

>> We thank the reviewer for this suggestion, and we now revisit Freire and critical pedagogy as part of the narrative of the results and discussion. We revisit those ideas when analysing the data including addressing how the students engaged in dialogue and detailing how the transformation and action was visible in terms of taking action and showcasing empathy. This is best captured in the extended paragraph at the start of the discussion (Line 390): *"The work through the series of workshops and engagements has effectively developed and deployed a methodology that can be best represented as a new climate praxis model, based on Freirean theory (Freire, 1970). This model builds a knowledge (gain), exploration (play) and action (Figure 7) framework, which combined connectivity to an environmental issue (coastal change) through to adaptation. The work was undertaken in partnership as educators and researchers, and we ensured that this was both age and socially/culturally-appropriate, building a 'critical dialogue' and recognising people's lived experiences. This asset is central to Freire's notion of praxis and forms an important shaping influence on the model. Reflecting on how this progressed through the methodology and the results, it is evident that this advanced through a double looping learning process and journey (Trajber et al., 2019), where a reflective approach within the cohort enabled participants to gain knowledge and understanding, explore and widen their perspectives through to beginning to act on coastal change within their broader community. In effect the work extends a climate literacy-based approach that seeks to enable students to become active participants and ensuring they are best prepared for the challenges that they face into the future with the knowledge to enable them to consider and derive potential solutions (Lawson et al, 2018; Hügel and Davies, 2020). Additionally, the interfaces between the principal components include looping and reflexivity, essentially involving each of the participants transiting their own "wave of change" (Jones et al. 2021). There is evidence in their journey as individuals and a collective of macro-level looping, where additional reflection leads to deeper knowledge, enhanced engagement and results in amplified action (Figure 7). Action in the work herein took the form of storytelling, the participants wanting to tell the narrative of their community to others and this was achieved through the production of their film."*

3. The Climate Praxis Model introduced is one of the most promising parts of the paper. It has real potential as a framework others could build on. But right now, it's a bit difficult to tell how the model emerged from the data itself. Was it built inductively based on what students did and said? Or was it designed in advance and then tested? Clarifying this could really help show the model's originality and relevance. Also, a brief discussion of

how the model relates to other educational frameworks—such as place-based learning, participatory education, or climate literacy—would ground it more firmly in existing scholarship and show where it makes a new contribution.

>> These details are now included in the discussion with addition of a paragraph that centrally addresses this point and the evolution of the model. This reads:

"The work through the series of workshops and engagements has effectively developed and deployed a methodology that can be best represented as a *new climate praxis model*, based on Freirean theory (Freire, 1970). The model was very much an evolution through the project and emerged from the engagements."

4. The paper could end on a stronger note by reflecting on what all of this means for climate education in disadvantaged or climate-vulnerable communities. How could this work be adapted elsewhere? What challenges might come up? What kind of support would schools need to do something similar?

>> An additional paragraph has been added to the conclusion to address this point and what it means for climate education in disadvantaged and climate-vulnerable communities. Notably we address the question R2 has concerning how the work could be adapted for elsewhere. This section now reads (Line 555): *"Participants shifted from initial disengagement and misconceptions to an empowered understanding of their role in addressing climate challenges. This offers critical insights into how climate education can be made meaningful and transformative for disadvantaged or climate-vulnerable communities by centring local relevance, emotional engagement, and creative expression. The Climate Praxis Model demonstrates that even in communities that are initially disengaged from climate issues, young people can become powerful agents of change when education is responsive to their lived experiences and cultural context. Adaptation of this model elsewhere would require a strong commitment to place-based learning, teacher training in emotionally responsive pedagogies, and institutional flexibility to move beyond standardised curricula and deficit-driven models. Key challenges may include educator confidence in facilitating open-ended creative work, and the need for localised climate contexts and community partnerships. However, as the success of this project has shown, such investment can yield not only deeper climate literacy but also foster the empathy, hope, and agency needed for collective climate action, particularly in communities at the frontline of environmental change."*

5. While Freire is used for critical pedagogy, the paper sometimes relies on him a little symbolically. A more grounded use of his work could enrich the analysis. Also, adding more recent references in climate education would situate the work more firmly in the field and help readers connect it with similar efforts globally.

>> The paper does rely on, and is inspired by Freire. This has been made clearer in a few places in the text. To address the key point of R2 here additional contextual references to climate education have been added to elements of the introduction and discussion.

---

## Author Response (AR2)

We thank the editor for the careful consideration of our revised manuscript and address the remaining issues flagged to us. We detail the changes we have made to the manuscript below:

1. We have edited the start of the discussion in order to better frame the section and the need for additional consideration of theoretical background. This now reads:

   *"The results record the participant's journey as the programme of workshops progressed and they increased their knowledge of climate change and also engaged with wider environmental issues in their locale, along with their own place(s) (with)in their community. A set of themes emerged from our coding of the observations of the sessions and analysis of the creative materials produced. These themes and observations have a suite of implications concerning how to best engage at risk coastal communities who are facing the impacts of climate driven erosion. This includes results that indicate the need for a broader-based climate education, which critically has local context embedded in order to engender an understanding, and action, across intergenerational and community-based dialogues. This section highlights and critically analyses the themes emergent from the results, framing the work within a broader theoretical context."*

2. In Section 4.1, we have moved content from the introduction and the methodology and reworked this into a set of paragraphs in the discussion where the new model is presented. This ensures that the theoretical framing around the new Climate Praxis Model is evolved within the same section the new model is introduced, ensuring that the theoretical foundations of Freire and the critical considerations are central. This new section (Discussion 4.1) reads:

   *"The work, through the series of workshops and engagements, has effectively developed and deployed a methodology that can be best represented as a new climate praxis model, based on Freirean theory (Freire, 1970). As noted above the PAR sessions were grounded in an approach aligned to the theoretical framing of Freire's critical consciousness and thus an engagement with theories of knowledge co-production, knowledge democracy (Duncan-Andrade and Morrell, 2008; Chapman, 2019; Stern, 2019), and communities of practice (Lave & Wenger, 1991). The new Climate Praxis model emerged from this framing, contributing conceptual advances through integrating rights-based climate education with storytelling as a method of participatory inquiry and intergenerational dialogues. Similarly, the creative methodologies deployed across the workshops serve as tools of knowledge mobilisation and reposition youth as legitimate knowledge producers and policy actors, thus offering a replicable approach to community-led climate resilience. In this way, the new*

*Climate Praxis educational model offers both theoretical and practical advancements in participatory climate education, positioning it as a new conceptual tool for future research and action. Freire (1970: p52) notes that for a praxis to be realised 'the oppressed must confront reality critically, simultaneously objectifying and acting upon that reality' and adds that 'critical and liberating dialogue, which presupposes action, must be carried on with the oppressed at whatever the stage of their struggle for liberation' (p.65)."*

3. An additional paragraph has also been added to the conclusion section detailing the outcomes.

*"The outcomes of the activities demonstrate strong participant engagement, particularly in grappling with the implications of climate change. These findings have important consequences for effective geoscience communication and broader climate change education. We argue that the new Climate Praxis Model offers a valuable template for both. The model proved effective when evolved to the specific context of a disadvantaged, at-risk coastal community. In this setting, knowledge transfer and engagement served as critical first steps. The workshop-based learning process successfully scaffolded participants' understanding, empowering them to take informed, meaningful action."*